# Fermentation of cv. Kalamata Natural Black Olives with Potential Multifunctional Yeast Starters

**DOI:** 10.3390/foods11193106

**Published:** 2022-10-06

**Authors:** Stamatoula Bonatsou, Efstathios Z. Panagou

**Affiliations:** Laboratory of Microbiology and Biotechnology of Foods, Department of Food Science and Human Nutrition, School of Food and Nutritional Sciences, Agricultural University of Athens, Iera Odos 75, GR-11855 Athens, Greece

**Keywords:** black olives, yeasts, Kalamata olives, functional foods, fermentation

## Abstract

The purpose of this study was to explore the inoculated fermentation of cv. Kalamata natural black olives using selected strains of yeast cultures with multifunctional potential. For this purpose, five yeast starters belonging to *Candida boidinii* (four starters) and *Saccharomyces cerevisiae* (one starter), previously isolated from table olive fermentation of the same variety and screened for their technological characteristics and probiotic potential, were inoculated in brines at the beginning of fermentation. Microbial populations (lactic acid bacteria, yeasts, and *Enterobacteriaceae*), pH, titratable acidity, organic acids, and ethanol were monitored during fermentation for a period of 5 months. At the same time, the survival of each starter was assessed by culture-dependent molecular identification at the beginning (0 days), middle (75 days), and final stages (150 days) of fermentation in the brines and olives (at the end of the process only). The results revealed the coexistence of yeasts and lactic acid bacteria (LAB) throughout fermentation in most processes and also the absence of *Enterobacteriaceae* after the first 20 days of brining. The population of yeasts remained 2 log cycles below LAB counts, except for in the inoculated treatment with *C. boidinii* Y28, where the yeast starter prevailed from day 60 until the end of the fermentation, as well as in the inoculated treatment with *C. boidinii* Y30, where no LAB could be detected in the brines after 38 days. At the end of the process, LAB ranged between 4.6 and 6.8 log_10_ CFU/mL, while yeasts were close to 5.0 log_10_ CFU/mL, except for the inoculated fermentation with *C. boidinii* Y27 and spontaneous fermentation (control), in which the yeast counts were close to 3.5 log_10_ CFU/mL. At the end of fermentation, the recovery percentage of *C. boidinii* Y27 was 50% in the brines and 45% in the olives. *C. boidinii* Y28 and *S. cerevisiae* Y34 could be recovered at 25% and 5% in the brine, respectively, whereas neither starter could be detected in the olives. For *C. boidinii* Y30, the recovery percentage was 25% in the brine and 10% in the olives. Finally, *C. boidinii* Y31 could not be detected in the brines and survived at a low percentage (10%) in the olives.

## 1. Introduction

Table olives are fermented products of high nutritional value because of their main components, such as antioxidant compounds and unsaturated fatty acids. Table olive fermentation, despite its economic significance, is mainly craft-based and carried out empirically by the table olive industry. Fermentation is undertaken spontaneously through the synergistic action of the autochthonous microbiota of the olives, namely, LAB, yeasts, and *Enterobacteriaceae*, depending on various intrinsic and extrinsic factors [1]. LAB provide microbiological stability to the final product through brine acidification by the production of lactic acid from fermentable substrates, while yeasts contribute to the determination of the sensory profile of the final product through the production of desirable metabolites and volatile compounds. The use of microorganisms related to table olives with multifunctional potential as starters for fermentation may reduce the risk of spoilage and contribute to a more controlled and reproducible fermentation process [2,3,4,5,6,7]. Moreover, the use of starter cultures has been shown to result in a significant decrease in the fermentation time, as well as the production of an improved sensory profile of the final product [5,8]. The selection and implementation of a starter culture is a complicated procedure. Microorganisms must fulfill specific criteria in order to be selected as starter culture candidates. Specifically, the main prerequisites include adaptation to the environment of fermentation due to acidified and high-salt-concentration brines, dominance over the indigenous microbiota, and the ability to attach to the olive epidermis and form biofilms. Furthermore, apart from the above-mentioned features, starter cultures should survive in conditions simulating digestion in adequate populations and have the ability to adhere to and colonize the epithelial cells in the intestine, providing both antimicrobial activities against pathogens and improvement in intestinal function through the production of specific enzymes. The ability of some strains to produce the enzyme β-glucosidase leads to the degradation of oleuropein and the natural debittering of olives without the use of chemical compounds. In addition to this enzyme, several studies have shown the ability of some strains to produce lipolytic enzymes, such as esterases and lipases, which contribute to the aroma formation through the metabolism of fatty acids, as well as phosphatases (acid and alkaline), which are mainly responsible for the release of inorganic phosphoru” fro’ phytic complexes and its availability to the cells. For all these reasons, the interest in the development and application of starters in table olive processing has increased over the past years [8,9,10,11,12,13].

LAB have been extensively studied for their multifunctional traits and their potential use as starters in table olive processing. However, since studies have shown that, in natural black olive fermentation, LAB are partially or completely inhibited because of the presence of phenolic compounds, low pH, and high salt concentration [14], yeasts could become the dominant microorganisms driving the fermentation process [8,15,16]. Several yeast species are well known for their important role in the production of fermented products and beverages, especially members of the genus *Saccharomyces*. These microorganisms can produce several compounds, such as ethanol, and thus contribute to the flavor of the final product [17,18]. Several authors have studied the beneficial properties of yeasts associated with table olives regarding their potential use as starters in natural olive fermentations [19,20,21,22,23,24,25,26]. The isolation and identification of table olive yeast microbiota using different processing styles revealed the high abundance of *Candida boidinii* and *Saccharomyces cerevisiae* species during different stages of fermentation. These species can survive the acidified conditions that occur during fermentation, colonize the olive surface, and positively affect the organoleptic profile of the final product [27,28,29,30,31,32,33,34]. In addition, specific strains of *C. boidinii* and *S. cerevisiae* presented high survival rates during the simulation of gastric and pancreatic digestion, high hydrophobicity and auto-aggregation ability, positive reactions for the enzymes esterase, β-glucosidase, and acid phosphatase [15,22], and a high ability to degrade cholesterol and form biofilms when co-cultured with LAB [20]. Furthermore, strains of *C. boidinii* presented strong lipase activity [35], whereas specific strains of *S. cerevisiae* showed the ability to produce killer toxins against human pathogens and spoilage microorganisms of table olives [36]. Based on the above, the aim of this study was to evaluate the inoculated fermentation of cv. Kalamata natural black olives with five strains of yeast starter cultures with multifunctional potential belonging to *C. boidinii* and *S. cerevisiae* and define their survival at selected time points of the process.

## 2. Materials and Methods

### 2.1. Olive Samples and Fermentation Procedures

Natural black table olives cv. Kalamata were harvested in early November to be processed according to the traditional anaerobic Greek-style method. The raw material was kindly provided by the table olive company Konstantopoulos S.A. located in Northern Greece and transported within 24 h to the Agricultural University of Athens. Fermentation was undertaken in plastic vessels (8 L) containing ca. 5.0 kg of olives and 3.0 L of freshly prepared 7.0% (*w*/*v*) NaCl brine acidified with 0.5% (*v*/*v*) vinegar (*ca.* 6.0%, *v*/*v*, acetic acid). Table olives were inoculated in monoculture with four strains of *C. boidinii* (Y27, Y28, Y30, and Y31) and one strain of *S. cerevisiae* (Y34) during the first day of fermentation to a final concentration of 3 × 10^6^ CFU/mL.

All treatments were performed in duplicate, and the fermentation took place at room temperature for an overall period of 150 days. Finally, during the process, coarse salt was added periodically to the brines to maintain the salt concentration at 7.0%. Specifically, dry salt was added to the brines on the 2nd day of fermentation and once per week for the first 3 weeks of the process. The strains employed in this work have been previously isolated during the spontaneous fermentation of the same table olive variety and selected for their technological features and probiotic potential. More specifically, strains *C. boidinii* Y28 and *S. cerevisiae* Y34 showed positive reactions for alkaline phosphatase, a hydrolytic enzyme responsible for the removal of a phosphate group from various types of molecules and the release of inorganic phosphorus, and were able to produce the enzyme β-glucosidase, which can hydrolyze oleuropein and remove the natural bitterness of olive drupes [16,37]. All selected strains showed lipolytic activity (positive reaction to esterase and lipase enzymes), a desirable characteristic for the enhancement of the olives’ flavor through the formation of several compounds through the metabolism of free fatty acids [22,25,35,38,39]. All of the selected starters showed high percentages of hydrophobicity, auto-aggregation, and overall survival (more than 65%) at the end of digestion, with the exception of Y28 (no survival). Strains 27, 30, and 31 showed low adhesion ability to Caco-2 cells, while none of the selected strains was hemolytic.

### 2.2. Inoculum Preparation

Starters were revived according to Blana et al. [40] with slight modifications. The five strains were grown in Yeast Mold (YM) broth medium (yeast extract 0.3% *w*/*v*, malt extract 0.3% *w*/*v*, bacteriological peptone 0.5% *w*/*v*, D-glucose 2% *w*/*v*, and agar 1.5% *w*/*v*) and incubated statically at 28 °C for 24 h. Working cultures were prepared by adding 100 μL of each strain to 100 mL of YM broth supplemented with 7.0% (*w*/*v*) NaCl and 0.5% (*v*/*v*) vinegar to simulate the conditions at the beginning of fermentation according to local practice. The final inoculum was also prepared according to Blana et al. [40] after centrifugation and pellet resuspension in sterile Ringer solution. The population of each inoculum was estimated by plate counting: it was 7.0 log CFU/mL for *C. boidinii* Y27, 7.5 log CFU/mL for *C. boidinii* Y28, Y30, and Y31, and 8.0 log CFU/mL for *S. cerevisiae* Y34. The final volume of the inoculums added to the fermentation vessels was 900 mL for *C. boidinii* Y27, 300 mL for *C. boidinii* Y28, Y30, and Y31, and 90 mL for *S. cerevisiae* Y34 to achieve an initial yeast population in the brines of ca. 6.0–7.0 log CFU/mL.

### 2.3. Microbiological Analyses

Microbiological analyses were performed to determine the population dynamics of LAB, yeasts, and *Enterobacteriaceae* in the brines throughout fermentation and in the olives (at the end of fermentation only). For this purpose, 1 mL of brine was aseptically transferred to 9 mL of quarter-strength Ringer’s solution. In the case of olives, 10 g of olive pulp was aseptically removed by means of a sterile scalpel and forceps from four olives, taken from different depths of the vessel, added in 90 mL of the same Ringer’s solution, and homogenized in a Stomacher device (LabBlender, Seward Medical, London, UK) at room temperature for 60 s. Decimal dilutions were prepared with the same diluent, and duplicate 1.0 or 0.1 mL samples of the appropriate dilutions were mixed or spread on the following media: (i) de Man-Rogosa-Sharpe (MRS, 401728, Biolife, Milan, Italy) adjusted to pH 5.7 and supplemented with 0.05% (*w*/*v*) cycloheximide (AppliChem GmbH, Darmstadt, Germany) for the enumeration of LAB, incubated at 25 °C for 72 h, (ii) Rose Bengal Chloramphenicol agar (RBC, supplemented with selective supplement X009, Bury, UK) for the enumeration of yeasts, incubated at 25 °C for 48 h, and (iii) Violet Red Bile Glucose agar (VRBGA, Biolife, Milan, Italy) for the enumeration of *Enterobacteriaceae*, incubated at 37 °C for 24 h. The results are expressed as log values of colony-forming units per mL (log CFU/mL) or g (log CFU/g) in the case of brine and olives, respectively.

### 2.4. Yeast Inoculation and DNA Amplification

Yeast colonies were selected from RBC plates according to [41] and purified by successive streaking on YM medium. Pure cultures were maintained at −80 °C in the same medium supplemented with 20% glycerol. Yeast species diversity was evaluated in the brines at three different time points, namely, at the beginning (day 0), middle (day 75), and end of fermentation (150 days), and in the olives at the end of fermentation. A total of 400 isolates (20 colonies selected per strain × 5 strains × 3 sampling points for brines and 20 colonies per strain × 5 strains at the end of the process for olives) were subjected to PCR (ProFlex PCR System, Applied Biosystems, Foster City, CA, USA) with the oligonucleotide primer (GTG)_5_ after DNA extraction for yeast cell lysis, performed according to Bonatsou et al. [15] with slight modifications. The survival of the starters was evaluated by comparing the profiles obtained by gel electrophoresis (GT Cell, Bio Rad, Hercules, CA, USA) in 1.5% agarose at 100 V for 1.5 h with Bionumerics software version 6.1 (Applied Maths, Sint-Martens-Latem, Belgium). The results are expressed as survival percentages, indicating the number of isolates with identical profiles to the yeast starter per the total number of 20 isolates per inoculated fermentation and sampling time. 

### 2.5. Physicochemical Analyses

The changes in pH, titratable acidity, and salt concentration in the brines were monitored throughout fermentation according to Garrido-Fernández et al. [42]. Organic acids (lactic, acetic, malic, citric, tartaric, and succinic) and ethanol were determined by HPLC, as detailed elsewhere [21]. All analyses were performed in duplicate, and the results are expressed as mean values ± standard deviation.

### 2.6. Data Analysis

Data exploration and interpretation were based on multivariate statistical analysis. The input matrix for the analysis contained the counts of the different microbial groups enumerated during fermentation, as well as the values of pH, titratable acidity, the profile of organic acids, and ethanol. Prior to analysis, the data were autoscaled to avoid bias due to differences in scale. Hierarchical Cluster Analysis (HCA) was employed to explore the relationships between the variables and the inoculated fermentation procedures. HCA was performed based on the Euclidean distance as a similarity measure and Ward’s linkage as a clustering algorithm. The outcome of the analysis was graphically illustrated in the form of a heatmap using Metabo-Analyst software ver. 3.0 [43]. In addition, Principal Component Analysis (PCA) was performed using the Pearson correlation matrix of the variables. The purpose of the analysis was to project the initial variables into the subspace defined by the reduced number of principal components in order to identify any correlations among them. PCA analysis was performed using the software Statistica ver. 7.1 (StatSoft Inc., Tulsa, OK, USA).

## 3. Results and Discussion

### 3.1. Population Dynamics during Fermentation

The population dynamics of the enumerated microbial groups in the brines during fermentation are presented in Figure 1. The changes in the population of *Enterobacteriaceae* were very similar between all fermentations. Specifically, they ranged from 2.1 to 3.9 log CFU/mL at the beginning of fermentation and presented an increase of ca. 1.0–1.5 log units during the first 8–12 days, followed by a rapid decrease thereafter, and could not be detected after 20 days of fermentation. Regarding yeasts, their population was close to 2.0 log CFU/mL in the spontaneous process (control) and ranged from 5.0 to 6.5 log CFU/mL among the inoculated treatments at the onset of fermentation. Yeasts coexisted with LAB in all processes, and their population was maintained 2 log cycles below LAB counts in all cases, with the exception of the inoculated treatment with *C. boidinii* Y28, where yeasts became the dominant microbial group after 60 days of fermentation. It needs to be noted that in this particular fermentation, LAB coexisted with yeasts, but their populations were maintained at 1.0–1.5 log CFU/g lower until the end of the process. Another noteworthy observation is that in the inoculated fermentation with *C. boidinii* Y30, the population of LAB increased rapidly within the first 12 days and reached a maximum of 7.3 log CFU/mL. However, a rapid decrease was observed after this point until day 38, at which point no LAB could be enumerated in the brines. Overall, at the end of fermentation, LAB were the dominant group, except for in the inoculated process with *C. boidinii* Y28 and *C. boidinii* Y30, with counts ranging between 4.6 and 6.8 log CFU/mL. In addition, the population of yeasts was 5.1 log CFU/mL on average (range: 3.8–5.9 log CFU/mL) in inoculated fermentations and 5.3 log CFU/mL in the spontaneous process.

The inoculated fermentation of table olives with yeast starters either as a single culture or in co-culture with LAB has been reported in the literature for both green and black olive varieties. Specifically, the yeast *Candida diddensiae* was used as a starter in cv. Arbequina natural green olives combined with/without a strain of *L. pentosus* [11]. It was reported that in the single-inoculation process, the yeast was able to reduce the survival period of *Enterobacteriaceae* compared with the spontaneous process, although it failed to colonize the brine until the end of fermentation. This observation was further confirmed in this research, where this microbial group could survive in the brines for 20 days in spontaneous fermentation, whereas, in the inoculated fermentations with the selected yeast starters, the survival period could be shortened by 4–6 days. This could be attributed to the fact that yeasts under stress conditions or in response to certain compounds contained in the fermentation medium (e.g., phenolic compounds) can synthesize bioactive compounds that are potentially toxic to other microorganisms [44]. In another work [45], a combined inoculum of *L. pentosus* and *S. cerevisiae* was employed in the fermentation of cv. Taggiaska black olives at different temperatures (23, 30, and 37 °C). The authors did not evaluate single-inoculum fermentation with the yeast species, as the presence of LAB is necessary to provide complete fermentation in terms of final pH and acidity values. Although the results of this work indicate successful inoculated fermentation, no microbiological analyses were undertaken throughout the process, and thus, the survival of the inoculated LAB and yeast cultures was not reported. Later, the potential for biofilm formation on the surface of cv. Conservolea black olives during inoculated fermentation with a combined inoculum of L. pentosus and Pichia membranifaciens was elucidated [46]. Both microorganisms exhibited in vitro probiotic potential, and they were used in the process in order to provide a functional food. The results indicated that the yeast species could survive in a high percentage at the onset of fermentation, but it could not survive in the final stage of the process (153 days). A sequential starter-driven fermentation process was also investigated for cvs. Kalamata and Conservolea black olives using an inoculum of *S. cerevisiae*/*L. mesenteroides* and *Debaryomyces hansenii*/*L. plantarum*, respectively [8]. The brines were first inoculated with the yeast species, followed by LAB after 63 days from yeast inoculation, a time point that coincided with the end of yeast fermentation. The results indicated successful fermentation in both varieties. The counts of yeasts in the inoculated process did not present considerable changes throughout fermentation and were maintained at ca. 5.5–5.8 log CFU/mL at the end of the process, which is in line with the final yeast counts observed in this work. The combination of the same yeast/LAB cultures was used in a different inoculation strategy for the same table olive cultivars [5].

In this work, three different inoculation approaches were employed, namely, the sequential inoculation of LAB followed by yeasts, the sequential inoculation of yeasts followed by LAB, and inoculation with a combined culture of LAB and yeasts. The results showed that the yeast population at the end of fermentation ranged from ca. 5.0–6.0 log CFU/mL for both cultivars, which coincides with the range of the final counts of yeast starters reported in our work. The final populations of yeasts and LAB observed in this work are also in good agreement with a recent work [24] that investigated the efficacy of yeast starters in controlling the fermentation of cv. Kalamata black olives in pilot-scale fermentations. The olives were subjected to inoculated fermentation using the Greek method using a strain of *S. cerevisiae* isolated from a previous fermentation of the same cultivar and also a commercial *S. cerevisiae* baker’s yeast. The authors reported high yeast populations (ca. 5.0–6.7 log CFU/mL) in all inoculated samples at the end of fermentation, as well as high levels of indigenous LAB (ca. 4.9–5.7 log CFU/mL). Finally, in a recent work [4], cv. Taggiasca black table olives were subjected to industrial-scale processing in different brine solutions (8 and 12%, *w*/*v*, NaCl) using the following yeast species as single cultures: *Candida adriatica*, *C. diddensiae*, *Cyteromyces matritensis*, *Nakazawaea molendiniolei*, *Saccharomyces cerevisiae*, and *Wickerhamomyces anomalus*. The authors reported that the salt concentration affected the survival of the inoculated yeasts. Thus, at the end of fermentation (120 days), *N. molendiniolei* and *C. matritensis* could not survive in high-salt brines, whereas *W. anomalus*, *C. diddensiae*, and *C. adriatica* presented high survival and dominated over the indigenous yeast microbiota.

### 3.2. Physicochemical Changes in the Brines during Fermentation

The values of pH after 1 day of brining ranged from 4.45 to 5.56 in the inoculated fermentations, whereas in the spontaneous process (control), a pH value of 6.17 was measured (Figure 2A). A similar pattern of pH values was observed in all fermentations, with a gradual decrease until day 12 that was dependent on the yeast strain employed in the process. From this time onwards, the values of pH reached a plateau until the end of fermentation. The lowest pH values at the end of the process were measured in the inoculated fermentations with *C. boidinii* Y27 and *C. boidinii* Y31 (pH 3.77), followed by *C. boidinii* Y28 with pH 4.15. The other treatments presented pH values above 4.3, which is considered the maximum pH value for natural olives according to the Quality Management Guide for the table olive industry of the IOC [47], namely, 4.49 (control treatment), 4.61 (*C. boidinii* Y30), and 5.03 (*S. cerevisiae* Y34).

Moreover, the changes in titratable acidity were in good agreement with the changes in pH. Specifically, according to Figure 2B, the highest acidity (0.815 g lactic acid/100 mL brine) was obtained after 150 days in the inoculated fermentation with *C. boidinii* Y27, followed by *C. boidinii* Y31 (0.697 g lactic acid/100 mL brine), whereas the remaining fermentations presented acidity values that ranged from 0.541 g lactic acid/100 mL brine (*C. boidinii* Y28) to 0.343 g lactic acid/100 mL brine (*C. boidinii* Y30). These values were higher than the minimum value of titratable acidity (0.3 g lactic acid/100 mL brine), also reported in the trade standard of the IOC for natural olives that are preserved with their specific chemical characteristics attained during fermentation [48].

Our results are comparable to those of Ciafardini and Zullo [4], who performed single-culture-inoculated fermentations of cv. Taggiasca black olives with selected species of yeasts and reported that the final pH and titratable acidity values attained in 8% salt brines, which is comparable to the 7% salt level used in this work, ranged from 4.33–4.36 and 0.488–0.524 g lactic acid/100 mL brine. However, slightly lower final pH values (4.21 to 4.27) were reported for cv. Kalamata black olives during sequential inoculation with selected yeast and LAB cultures [5]. This was attributed to the acidification caused by the presence of the added LAB starter cultures in the brines during the sequential inoculation strategy. Finally, similar minimum pH values in the range of 4.2–4.3 were reported for inoculated pilot-scale fermentations of cv. Kalamata black olives [24] with *S. cerevisiae*, indicating the contribution of the indigenous LAB microbiota in the process.

The changes in the concentrations of organic acids and ethanol in the brines are shown in Figure 3. As expected, due to the absence of LAB from the inoculated fermentation with *C. boidinii* Y30 after day 40, the concentrations of the organic acids were lower compared to the other treatments, where LAB dominated throughout fermentation. Citric acid was the main acid detected, followed by malic and lactic acids, which showed a gradual increase until day 16 and then remained stable until the end of the process. For all other treatments, lactic acid was the main acid with considerable presence in the brines, followed by malic and citric acids [8,21]. In all treatments, succinic acid was present in lower concentrations not exceeding 2.0 g/L, whereas tartaric acid was not detected. The detection of acetic acid in low concentrations (0.5–1.0 g/L) could be attributed to both yeast and LAB activity, which can produce acetic acid through the assimilation of citrate, as reported by several authors in previous studies [44,49,50,51]. Ethanol is the main metabolite produced by the yeasts and highly contributes to the sensorial profile of the final product [6,52]. Its concentration ranged from 0.5 to 1.0 g/L in most treatments, showing the highest values, close to 2.0 g/L, in fermentations with *C. boidinii* Y30 and *S. cerevisiae* Y34.

### 3.3. Survival of the Inoculated Yeast Strains

The survival percentages of the five yeast starters for each fermentation and sampling time are presented in Table 1, whereas the cluster analysis of yeast isolates is presented in Appendix A. The comparison of the electrophoretic profiles from cluster analysis showed that the recovery of the inoculated strains in the brines on day 0 was 85% for *C. boidinii* Y30 (17/20 isolates), 90% for starter *C. boidinii* Y31 (18/20 isolates), and 100% for starters *C. boidinii* Y27, *C. boidinii* Y28, and *S. cerevisiae* Y34, indicating the successful inoculation of the brine at the beginning of fermentation. These results are in good agreement with previous reports on the inoculated fermentation of cv. Kalamata black olives with yeast starters, reporting high recovery rates (>70%) of the inoculated cultures at the beginning of fermentation [5,8]. Lower recovery rates were obtained on day 75, where the survival percentages in the brines were 5% for *S. cerevisiae* Y34 (1/20 isolates), 10% for *C. boidinii* Y28 (2/20 isolates), 15% for *C. boidinii* Y31 (3/20 isolates), 30% for *C. boidinii* Y30 (6/20 isolates), and 45% for *C. boidinii* Y27 (9/20 isolates). At the end of fermentation (day 150), the survival rate of *C. boidinii* Y27 was 50% (10/20 isolates) in the brines and 45% (9/20 isolates) in the olives. For *C. boidinii* Y28, the survival rate was 25% in the brines (5/20 isolates), while the yeast starter could not be recovered from the surfaces of olives. The survival of *C. boidinii* Y30 in the brines and olives was 25% (5/20 isolates) and 10% (2/20 isolates), respectively, while the yeast *C. boidinii* Y31 could be recovered only from the olives at 10% (2/20 isolates). Finally, for *S. cerevisiae* Y34, very low survival rates were obtained in the brines (5%, 1/20 isolates), whereas the starter was not recovered from the olives.

The survival rates obtained at the end of fermentation could not support the establishment of dominance by the selected yeast starters in the fermentations. The only exception was *C. boidinii* Y27, presenting the highest survival rates in both the brines and olives, making this yeast a promising starter culture for cv. Kalamata black olive fermentation, thus enhancing the multifunctional profile of the final product. Similar survival rates of *C. boidinii* Y27 have been reported in the fermentation of the same table olive cultivar with the strain *S. cerevisiae* KI 30-16 following a sequential inoculation strategy with *Lc. mesenteroides* K T5-1, establishing a 60% survival rate at the end of fermentation (90 days) [8]. In a recent work [5], the same yeast/LAB starter cultures were used in the inoculated fermentation of cv. Kalamata black olives following diverse inoculation strategies, including the combined inoculation of the two microorganisms at the onset of the process, as well as sequential inoculation, initially with the yeast strain, followed by the LAB culture after 63 days of fermentation. The authors reported that at the end of fermentation (105 days), the recovery rates of the yeast starter were 50% and 40%, respectively. In a previous work [4], cv. Taggiasca black olives were fermented with selected yeast species as starter cultures in different salt brines and reported isolation frequencies of 40%, 42%, and 45% for *C. adriatica*, *W. anomalus*, and *C. diddensiae*, respectively, in 8% salt brines at the end of fermentation (120 days). In another work [46], cv. Conservolea black olives were fermented using a combined inoculum of multifunctional starters, namely, *L. pentosus*, and *P. membranifaciens.* The authors reported that the recovery rate of yeast from the surfaces of olives was 100% after one day of inoculation, but the strain could not be recovered at the end of fermentation (153 days). The difficulty of some yeast strains to survive in high numbers and finally dominate in the brines could be attributed to competition with the already-existing indigenous LAB/yeast microbiota in the process [5].

Another important issue is that certain yeast species have the ability to auto-aggregate and form biofilms on the surfaces of olives [6,18]. As a consequence, the survival rates of the yeast starter cultures should be determined not only in the brines but in the olives as well, because differences may occur, as illustrated in Table 1, which shows that higher survival rates were obtained in the brines compared to olives. This observation is important, given that the fermenting brine is discarded and the olives are packed in freshly prepared brine prior to marketing. Thus, it is necessary to know the survival rate of the inoculated starter culture attached to the surfaces of olives, especially when the selected starter culture, either yeast or LAB, is characterized by multifunctional potential and is employed in the fermentation in order to enhance the nutritional role of olives.

### 3.4. Exploratory Data Analysis of the Fermentation Profiles

Experimental data are presented in the form of a heatmap (Figure 4), a two-dimensional graphical illustration in which physicochemical and microbiological characteristics are individually depicted by a single row, while the different fermentations are represented in columns. Various shades of brown and blue colors in the heatmap correspond to strong and weak correlations between the microbiological and physicochemical parameters measured and the different inoculated fermentation treatments. The heatmap shows the discrimination of the fermentations in two main clusters. The first cluster (Cluster I) included spontaneous fermentation (control) and the two inoculated fermentations with *C. boidinii* Y30 and *S. cerevisiae* Y34, while the second cluster (Cluster II) included the inoculated fermentations with *C. boidinii* Y27, *C. boidinii* Y28, and *C. boidinii* Y31. Moreover, physicochemical and microbiological parameters were classified into two main groups. The first group included the LAB and yeast populations, succinic and lactic acids, and titratable acidity, which were mainly correlated with Cluster II. The second group included propionic, citric, acetic, and malic acids, pH, and ethanol and was correlated with Cluster I.

The results of the PCA analysis showed that there were three principal components (PCs) with eigenvalues > 1.0 accounting for 46.29%, 19.41%, and 10.62% of the variation in the data set, indicating that the initial 12 variables could be expressed as a linear combination of 3 PCs explaining 76.32% of the total variance. Biplots displaying PC1 vs. PC2 are presented in Figure 5, illustrating the projection of the variables on the plane defined by the first two principal components. PC1 was related to the fermentation time (days), since there was a transition of the sampling points from the left axis of PC1, corresponding to a short fermentation time, to the right side of the axis, corresponding to a longer fermentation time (Figure 5B). PC2 could be related to the different fermentation processes. Specifically, the inoculated fermentations with *C. boidinii* Y30 and *S. cerevisiae* Y34 were located below the PC1 axis and on the lower left side of the plane. These fermentations were correlated with malic, citric, and acetic acids, as well as with ethanol (Figure 5A). In addition, the inoculated fermentations with *C. boidinii* Y27, *C. boidinii* Y28, and *C. boidinii* Y31 were located above the PC1 axis and on the upper left side of the plane, associated with lactic and succinic acids, acidity, and the population of LAB (MRS). The nearly 180° angle for the variable pH against the variables lactic acid, acidity, and LAB counts (MRS) indicates an inverse correlation, and thus, low pH values are due to high levels of lactic acid, acidity, and LAB population.

## 4. Conclusions

The present work investigated the inoculated fermentation of cv. Kalamata black olives with selected strains of *C. boidinii* and *S. cerevisiae*. All strains were previously characterized as having multifunctional properties, and thus, a prerequisite for a successful fermentation is the dominance of the inoculated culture at the end of fermentation. The presence of the yeast starters at the onset of fermentation had a positive impact on the reduction in the survival period of *Enterobacteriaceae* in the brines compared to spontaneous fermentation. However, strict monitoring of the process is required since some yeast strains may lead to high and non-acceptable pH values in the brines and therefore to a non-stable and potentially unsafe final product. In this context, the results obtained in this study showed that only *C. boidinii* Y27, *C. boidinii* Y28, and *C. boidinii* Y31 led to a final product with acceptable pH values that could ensure the microbiological safety of the final product.

For the other yeast starters, the pH values at the end of fermentation suggest that additional control measures should be undertaken to ensure the stability of the final product, such as acidification at the beginning of fermentation with acidulants. Concerning the survival of the inoculated yeast starters at the end of fermentation, only the strain *C. boidinii* Y27 could be recovered in satisfactory percentages in both brines and olives. Given the technological features and probiotic potential of this strain, it could be used as a starter culture either as a monoculture or in combination with a multifunctional LAB strain to produce functional black table olives. 

## Figures and Tables

**Figure 1 foods-11-03106-f001:**
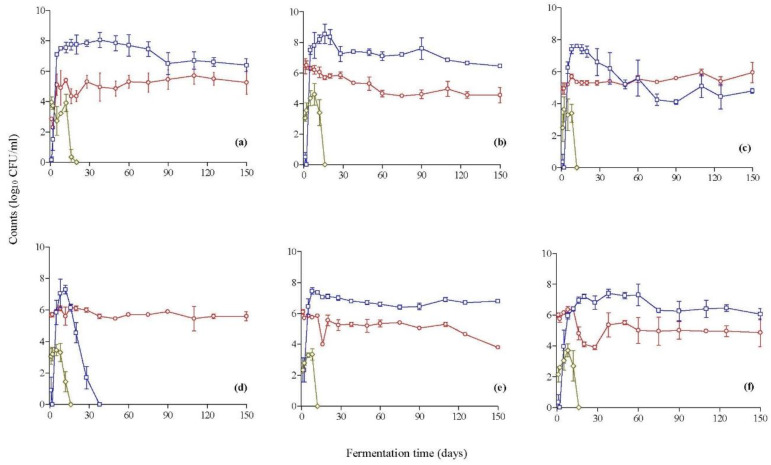
Changes in the population of LAB (**□**), yeasts (**◯**), and *Enterobacteriaceae* (**◊**) in the brines during the inoculated fermentation of cv. Kalamata natural black olives with yeast starters. (**a**) Spontaneous fermentation (control), (**b**) *C. boidinii* Y27, (**c**) *C. boidinii* Y28, (**d**) *C. boidinii* Y30, (**e**) *C. boidinii* Y31, and (**f**) *S. cerevisiae* Y34. Data points are mean values of duplicate fermentations ± standard deviation.

**Figure 2 foods-11-03106-f002:**
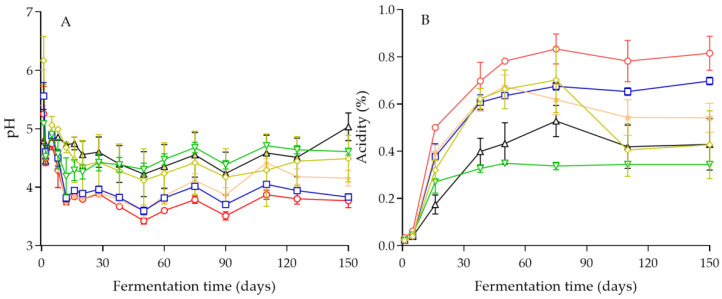
Changes in pH (**A**) and titratable acidity (%) (**B**) during inoculated fermentation of cv. Kalamata natural black olives with selected yeast starters: *C. boidinii* Y27 (**◯**), *C. boidinii* Y28 (★), *C. boidinii* Y30 (▽), *C. boidinii* Y31 (☐), *S. cerevisiae* Y34 (∆), and spontaneous fermentation (◊). Data points are mean values of duplicate fermentations ± standard deviation.

**Figure 3 foods-11-03106-f003:**
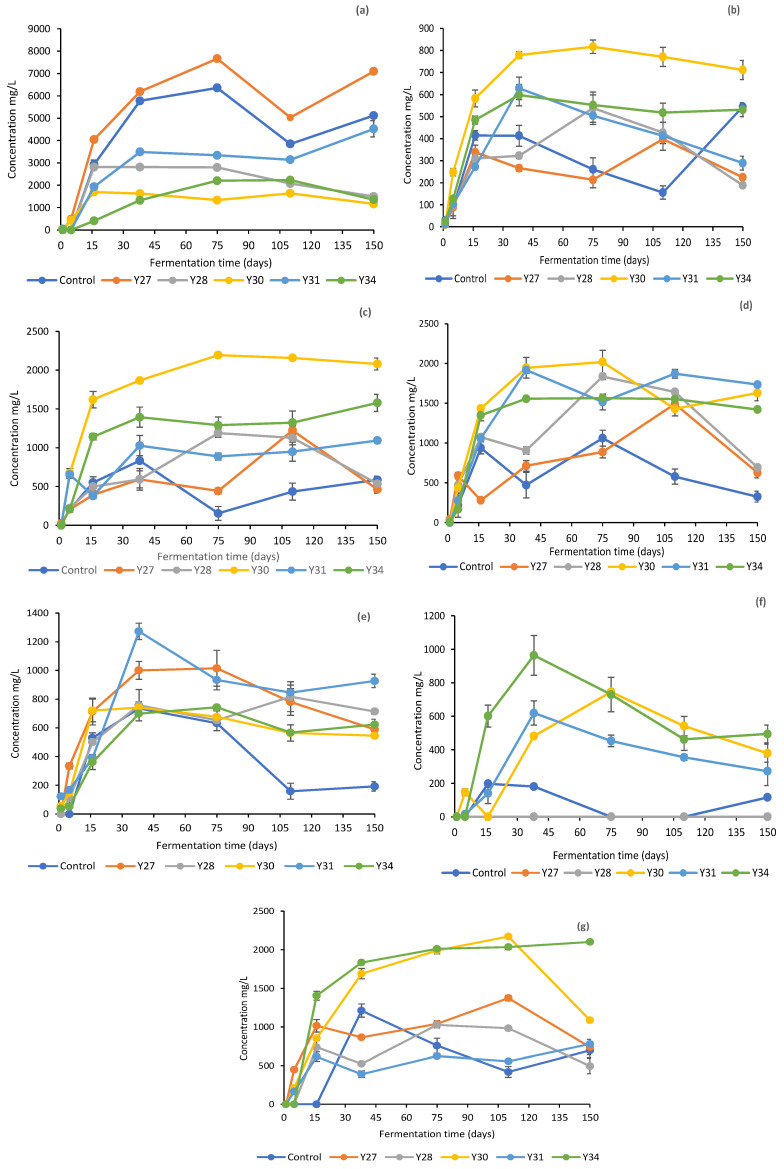
Changes in the concentrations of organic acids and ethanol in the brines during processing of cv. Kalamata natural black olives with selected yeast starters. (**a**) Lactic acid, (**b**) acetic acid, (**c**) citric acid, (**d**) malic acid, (**e**) succinic acid, (**f**) propionic acid, and (**g**) ethanol. Data points are mean values of duplicate fermentations ± standard deviation.

**Figure 4 foods-11-03106-f004:**
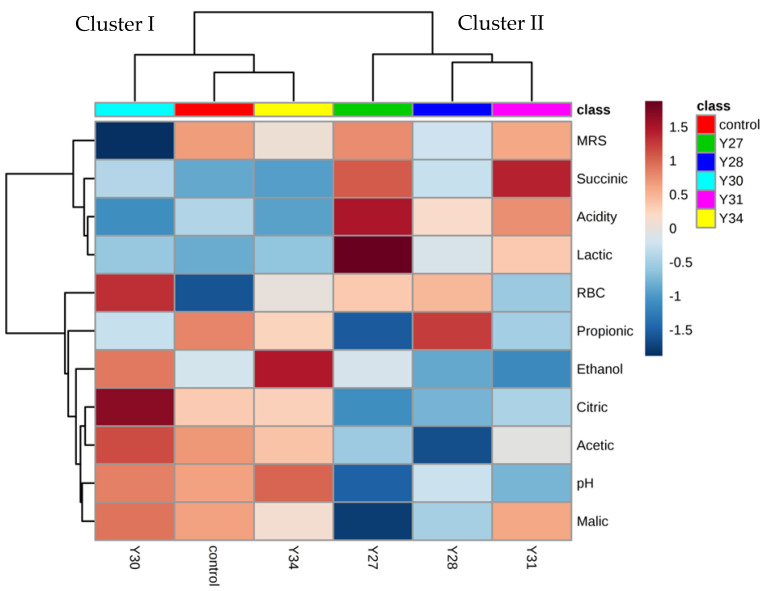
Hierarchically clustered heatmap of variables (LAB and yeast counts, organic acids, ethanol, pH, and acidity) and subjects (inoculated fermentations) (control: spontaneous fermentation; Y27, Y28, Y30, and Y31: inoculated fermentations with different strains of *C. boidinii*; Y34: inoculated fermentation with *S. cerevisiae.* MRS: lactic acid bacteria; RBC: yeasts).

**Figure 5 foods-11-03106-f005:**
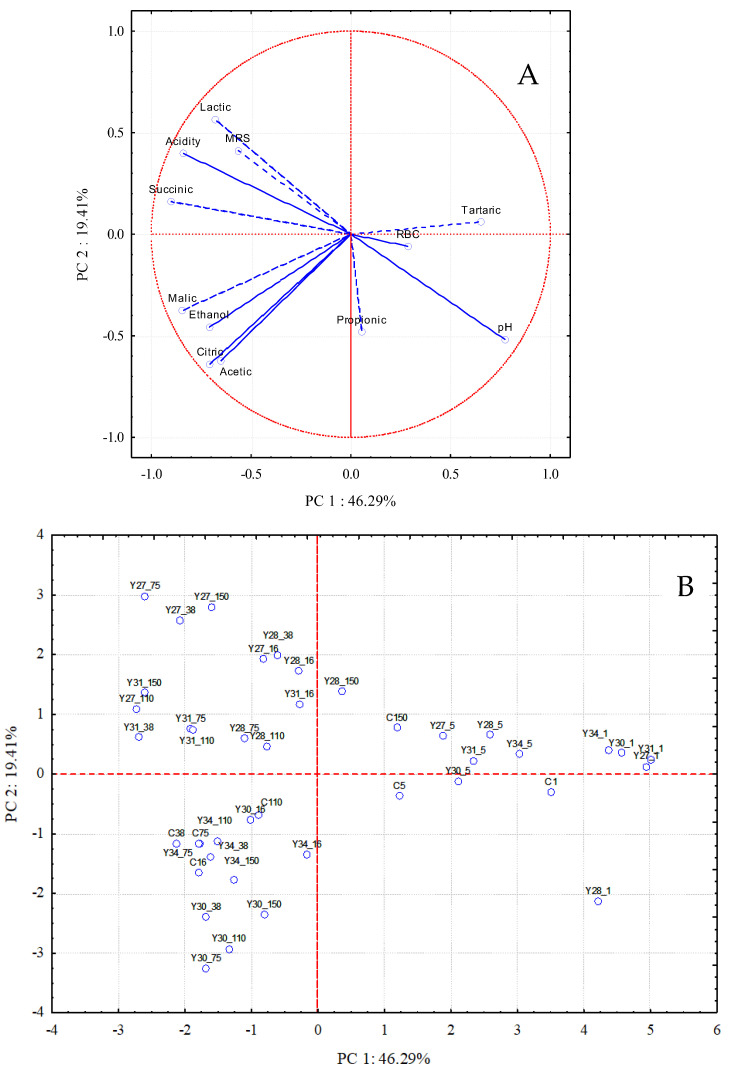
Plot of loadings: (**A**) microbiological and physicochemical variables and scores; (**B**) sampling days of the different fermentation processes in the plane of the first two principal components (C: spontaneous fermentation; Y27, Y28, Y30, and Y31: inoculated fermentations with different strains of *C. boidinii*; Y34: inoculated fermentation with *S. cerevisiae*; sampling time: 1, 5, 16, 38, 75, 110, and 150 days).

**Table 1 foods-11-03106-t001:** Starter culture survival rates (%) during different time points of fermentation processes.

Starter	Fermentation Time (Days)	Survival Rate (%)
*C. boidinii* Y27	0	100
	75	45
	150 ^brine^	50
	150 ^olives^	45
*C. boidinii* Y28	0	100
	75	10
	150 ^brine^	25
	150 ^olives^	0
*C. boidinii* Y30	0	85
	75	30
	150 ^brine^	25
	150 ^olives^	10
*C. boidinii* Y31	0	90
	75	15
	150 ^brine^	0
	150 ^olives^	10
*S. cerevisiae* Y34	0	100
	75	5
	150 ^brine^	5
	150 ^olives^	0

## Data Availability

Data are available on request to the corresponding author.

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
