# Peer review of "Fermentation of cv. Kalamata Natural Black Olives with Potential Multifunctional Yeast Starters"

_foods, 2022, doi:10.3390/foods11193106_

Round 1
Reviewer 1 Report
The manuscript describes the results of using potential multifunctional yeast starters for the fermentation of Kalamata natural black olives, paying particular attention to the survival of the diverse strains and species in the inoculum. The fermentation evolution was followed by analysing several physicochemical and microbiological parameters at selected time intervals. The composition of the olive biofilm was only checked at the end of the process. The different survivals of strains from the same species support the convenience of a detailed investigation before their inclusion in commercial starters. The manuscript is well structured and presents interesting results but sometimes describes results in excessive detail. A more general trend description will be acknowledged. Besides, some comments for the consideration of the authors follow.
Title. Reading it sounds a bit redundant. An alternative could be “Fermentation of cv. Kalamata natural black olives with potential multifunctional yeasts starters” or “Inoculation of cv. Kalamata natural black olives with potential multifunctional Candida boidinii and Saccharomyces cerevisiae starters”, which is more informative.
L21. the sentence “…in all fermentations, apart from…” sounds contradictory.
L25 and others (e.g. 214), similar to L21.
L 27-33. The paragraph is hard to follow. Please, try to facilitate understanding.
L217-230. Figure 1. The colour of the symbol for yeasts does not correspond to that in graphs.
L343- 362. Figure 3. The use of bar plots with this type of data is not recommended. Evolution could be preferably shown as in Figures 1 and 2.
L426-440. The explanation is not clear (e.g. line 429). Additionally, the figure legend does not explain some acronyms (MRS and RBC) in the heat map.
L465. Please consider changing the terms (“low values”)
L473. “above the line of origin”?
L519. “…the quality management guide for the table olive industry.”? Consider also that the Standard is provided for physicochemical characteristics of the packing brine.
Supplementary material. The figure size typos are minimal. Please enlarge them to make them readable.
Author Response
Comment 1: Title. Reading it sounds a bit redundant. An alternative could be “Fermentation of cv. Kalamata natural black olives with potential multifunctional yeasts starters” or “Inoculation of cv. Kalamata natural black olives with potential multifunctional Candida boidinii and Saccharomyces cerevisiae starters”, which is more informative.
Response: We agree with the reviewer to change the title of the manuscript. We have selected the first proposed title.
Comment 2: L21. the sentence “…in all fermentations, apart from…” sounds contradictory.
Response: The sentence has been modified to improve the meaning. Please see modified text.
Comment 3: L25 and others (e.g. 214), similar to L21.
Response: The wording has changed to improve the meaning.
Comment 4: L 27-33. The paragraph is hard to follow. Please, try to facilitate understanding.
Response: The paragraph has been modified to facilitate understanding.
Comment 5: L217-230. Figure 1. The colour of the symbol for yeasts does not correspond to that in graphs.
Response: The color for yeasts is red and the symbol is red circle. The color in the caption has changed to match with the graph.
Comment 6: L343- 362. Figure 3. The use of bar plots with this type of data is not recommended. Evolution could be preferably shown as in Figures 1 and 2.
Response: Figure 3 has been modified as requested.
Comment 7: L426-440. The explanation is not clear (e.g. line 429). Additionally, the figure legend does not explain some acronyms (MRS and RBC) in the heat map.
Response: The wording has been improved to clarify the meaning. Also the abbreviation of MRS and RBC has been added in the caption to clarify the meaning.
Comment 8: L465. Please consider changing the terms (“low values”)
Response: The sentence has been rephrased to improve the meaning.
Comment 9: L473. “above the line of origin”?
Response: Corrected. It was actually above and below PC1 axis.
Comment 10: L519. “…the quality management guide for the table olive industry.”? Consider also that the Standard is provided for physicochemical characteristics of the packing brine.
Response: we thank the reviewer for this comment. The text has been modified to improve the information provided.
Comment 11: Supplementary material. The figure size typos are minimal. Please enlarge them to make them readable.
Response: Typos were enlarged as suggested.
Reviewer 2 Report
The fermenting cv. Kalamata natural black olives belongs to table food with multifunctional potential, and which is very popular in Greece. In order to reveal the physicochemical characteristics and dynamic growth and decline characteristics of microorganism in brine and olive during the processing procedure of cv. Kalamata natural black olives, the investigation were implemented by inoculating in mono-culture with four strains of C. boidinii (Y27, Y28, Y30, and Y31) and one strain of S. cerevisiae (Y34) in table olives. At the same time, a number of indicators were examined, including microbial populations (lactic acid bacteria, yeasts and Enterobacteriaceae), pH, titratable acidity, organic acids and ethanol in different sampling time points during fermentation 5 months. In additional, except for Hierarchical Cluster Analysis (HCA) was employed to explore the relationship among the variables and the inoculated fermentation procedures for data analysis , the Principal Component Analysis (PCA) was performed using the Pearson correlation matrix of the variables.
As can be expected from the design of a single inoculation fermentation experiment, the fermentation products of the other four strains of yeast were not satisfactory except C. boidinii Y27, which had suitable pH and satisfactory recovery. Moreover, the mechanism by the C. boidinii Y27 inhibits the acid production of lactic acid bacteria needs to be further studied. In order to optimize the quality of fermentation products, it is necessary to further study the inoculation fermentation of mixed strains. But the study provides basic data for further research.
Based on the shortcomings existing in the current manuscript, it needs to make a major revision before accepting.
Other opinion
1. The line 45, the full stop are not marked; the line 83, there is excessive citation of literature.
2. The line108, the time node of periodically adding coarse salt is not specified. Whether the difference in salt addition could be a factor affecting microbial growth and metabolite aggregation?
3. Lines 134-135,the addition of strains should be more precisely controlled.
4. The line 144, lack of device information of " Stomacher device".
5. line 164,Lack of device information of "PCR".
6. line 166,Lack of device information of "gel electrophoresis".
7. lines 325-326, C. boidinii Y30 (), S. cerevisiae Y34 () in FIG. 2, lacks the polyline legend symbol.
8. lines 344-364,in fact,all analyses data of organic acids and ethanol did not show the symbol of the duplicate and statistical information according to the related paragraphs and Figure 3.
9. Table 1. The data of the survival rate lacked replicate and statistical processing information.
10. lines 469-471,It is not accurately expressed, after all, there are some points, such as spontaneous process (C) in 100 days beyond the scope.
11. Comparing the information in the figure,the information display of the sample time of 16 days is lacking in the title annotation of Figure 6.

Author Response
Comment 1: The line 45, the full stop are not marked; the line 83, there is excessive citation of literature.
Response: The missing full stop has been added in line 45. The number of citations in line 83 has been reduced as requested.
Comment 2: The line 108, the time node of periodically adding coarse salt is not specified. Whether the difference in salt addition could be a factor affecting microbial growth and metabolite aggregation?
Response: The time of salt addition is reported in the revised manuscript. The plan of salt addition was the same in all inoculated fermentations in order to keep the same salt concentration in all brines. There is no doubt that salt is a critical factor affecting microbial growth and metabolite aggregation. To cope with this issue and minimize the effect of different salt concentrations during the course of fermentation, the level of salt was maintained at the same level in all vessels.
Comment 3: Lines 134-135, the addition of strains should be more precisely controlled.
Response: The requested information about the addition and control of the starters has been added to the revised text.
Comment 4: The line 144, lack of device information of Stomacher device".
Response: The missing information has been added in the revised text.
Comment 5: line 164,Lack of device information of "PCR".
Response: The missing information has been added.
Comment 6: line 166,Lack of device information of "gel electrophoresis".
Response: The requested information has been added in the revised text.
Comment 7: lines 325-326, C. boidinii Y30 (), S. cerevisiae Y34 () in FIG. 2, lacks the polyline legend symbol.
Response: Corrected.
Comment 8: lines 344-364,in fact,all analyses data of organic acids and ethanol did not show the symbol of the duplicate and statistical information according to the related paragraphs and Figure 3.
Response: Figure 3 has been revised according to the comment raised by the first reviewer and the statistical information has been added in the revised caption. Also error bars were included in the graph.
Comment 9: Table 1. The data of the survival rate lacked replicate and statistical processing information.
Response: For the determination of the survival rate of the added starter cultures we were based on current practice on molecular biology and existing literature and pertinent studies performed by several authors, such as Tufariello et al. (2015, 2019), Chytiri et al. (2020), that focused on the survival of LAB and/or yeasts during the course of cv. Kalamata natural black olives.
Comment 10: lines 469-471,It is not accurately expressed, after all, there are some points, such as spontaneous process (C) in 100 days beyond the scope.
Response: This is true for the spontaneous process (C) where some points are scattered throughout the plane of PC1 and PC2. The sentence has been modified to improve the meaning.
Comment 11: Comparing the information in the figure, the information display of the sample time of 16 days is lacking in the title annotation of Figure 6.
Response: By mistake the time 16 was not included in the caption of Figure 6. This has now been corrected.
Round 2
Reviewer 2 Report
The author has responded to the proposed modification suggestions in a timely manner and made one-to-one modifications. However, the revised version still has the following problems:
1. The problem of excessive citation of literature has not been solved in the whole text . For example, lines100-102, the 8 literatures still used, lack of authoritative judgment of literature, and there is a simple stacking.
2. The introduction should be discussed in sections. Similarly, the level of paragraph recognition in the whole text needs to be improved.
3. The line numbers of the revised manuscript were arranged rough, resulting in the line numbers obscuring the graphic information.
4. The graphic format of Figure 3 should be consistent with that of Figures 1 and 2.
Author Response
Comment 1: The problem of excessive citation of literature has not been solved in the whole text. For example, lines100-102, the 8 literature still used, lack of authoritative judgment of literature, and there is a simple stacking.
Response: The number of references in lines 100-102 has been reduced as suggested by the reviewer.
Comment 2: The introduction should be discussed in sections. Similarly, the level of paragraph recognition in the whole text needs to be improved.
Response: Paragraphs have been created throughout the manuscript according to the suggestion of the reviewer.
Comment 3: The line numbers of the revised manuscript were arranged roughly, resulting in the line numbers obscuring the graphic information.
Response: This problem happened during the uploading of the document. We shall upload the file also as pdf document to avoid this issue.
Comment 4: The graphic format of Figure 3 should be consistent with that of Figures 1 and 2.
Response: We sincerely thank the reviewer for this comment. Figure 3 has been modified as suggested.